# The Neuroregenerative Effects of IncobotulinumtoxinA (Inco/A) in a Nerve Lesion Model of the Rat

**DOI:** 10.3390/ijms26157482

**Published:** 2025-08-02

**Authors:** Oscar Sánchez-Carranza, Wojciech Danysz, Klaus Fink, Maarten Ruitenberg, Andreas Gravius, Jens Nagel

**Affiliations:** 1Nonclinical Sciences & Operations, Merz Therapeutics GmbH, Eckenheimer Landstraße 100, D-60318 Frankfurt, Germany; oscar.sanchez@merz.de (O.S.-C.); andreas.gravius@merz.de (A.G.); 2Neurotoxin Biotechnology Development, Merz Therapeutics GmbH, Eckenheimer Landstraße 100, D-60318 Frankfurt, Germany; klaus.fink@merz.de

**Keywords:** neuroregeneration, pain, neuropathy, incobotulinumtoxin, botulinum toxin, BoNT/A, rat

## Abstract

The use of Botulinum Neurotoxin A (BoNT/A) to treat peripheral neuropathic pain from nerve injury has garnered interest for its long-lasting effects and safety. This study examined the effects of IncobotulinumtoxinA (Inco/A), a BoNT/A variant without accessory proteins, on nerve regeneration in rats using the chronic constriction injury (CCI) model. Inco/A was administered perineurally at two time points: on days 0 and 21 post CCI. Functional and histological assessments were conducted to evaluate the effect of Inco/A on nerve regeneration. Sciatic Functional Index (SFI) measurements and Compound Muscle Action Potential (CMAP) recordings were conducted at different time points following CCI. Inco/A-treated animals exhibited a 65% improved SFI and 22% reduction in CMAP onset latencies compared to the vehicle-treated group, suggesting accelerated functional nerve recovery. Tissue analysis revealed enhanced remyelination in Inco/A-treated animals and 60% reduction in CGRP and double S100β signal expression compared to controls. Strikingly, 30% reduced immune cell influx into the injury site was observed following Inco/A treatment, suggesting that its anti-inflammatory effect contributes to nerve regeneration. These findings show that two injections of Inco/A promote functional recovery by enhancing neuroregeneration and modulating inflammatory processes, supporting the hypothesis that Inco/A has a neuroprotective and restorative role in nerve injury conditions.

## 1. Introduction

Peripheral neuropathic pain is a debilitating chronic condition that affects up to 10% of the general population [1,2]. It can be caused by a wide range of conditions, including infections, genetic or metabolic diseases, and traumatic injuries, all of which result in lesions in the somatosensory system and nerve denervation [3]. Nerve damage leads to increased and spontaneous excitability of peripheral and central neuronal circuits, and nerve denervation can result in tremendous functional disability, particularly when nerve regeneration is slow or incomplete. Therefore, understanding the molecular and cellular mechanisms behind nerve regeneration after nerve injury and developing alternative treatments to accelerate axonal regeneration are of general interest due to their clinical challenges.

The use of Botulinum Neurotoxin type A (BoNT/A), produced by *Clostridium botulinum*, is investigated as an attractive treatment option to promote nerve regeneration and sprouting [4,5,6,7,8]. In different pre-clinical studies employing nerve lesion models, treatment with BoNT/A has been shown to enhance functional recovery, indicating nerve regeneration [5,6,7,8]. In particular, the attractiveness of this approach lies in its efficacy and its ability to be complemented by safe, transient follow-up injections, thereby reducing the burden of daily oral administrations [1,9]. Most of the conclusions from pre-clinical studies, however, are based on a single BoNT/A administration, with different administration timings, types of BoNT/A formulations, doses, and areas of injections. Despite differences in experimental designs, the available evidence indicates that in rodents, using various pain models, a single dose of BoNT/A not only provides analgesic effects but also enhances functional recovery. The latter is accompanied by accelerated nerve regeneration and axonal remyelination facilitated, in part, by the dedifferentiation and proliferation of Schwann cells [5,6,8,10]. Additionally, it has been proposed that BoNT/A contributes to nerve regeneration by inhibiting the inflammatory process occurring post nerve injury [6,11]. However, nerve regeneration has not been investigated with repeated applications of BoNT/A. This is of particular importance because repeated applications have been suggested to achieve significantly stronger effects against peripheral neuropathic pain than a single application [1]. Moreover, the effect of BoNT/A on the inflammatory process post nerve injury has not been studied yet. Here, we have investigated the effect of IncobotulinumtoxinA (Inco/A), a purified form of BoNT/A without complexing proteins [12], on nerve regeneration in rats using the chronic constriction injury (CCI) pain model.

## 2. Results

### 2.1. Inco/A Accelerated Functional Recovery in Rats Post CCI

The Sciatic Functional Index (SFI) is a functional assessment that provides information about recovery of the sensory–motor connections related to gait function mediated by the sciatic nerve in animal models [13,14]. We evaluated the SFI in rats post CCI in the absence or presence of Inco/A. As shown in Figure 1, sham rats did not develop any motor deficits during the three time points measured. However, rats subjected to CCI showed a robustly reduced SFI compared to sham animals already on day 3 post-surgery (Figure 1b,c). In vehicle-injected rats, this reduction was maintained until day 56 post CCI, showing a sequential and partial recovery during the time course (Figure 1c). Interestingly, animals treated with Inco/A did not show any functional recovery on day 3 post CCI, suggesting that perineural Inco/A administration did not accelerate nerve recovery at this time point in rats. However, on day 35 post nerve injury, rats showed an absolute 20% SFI reduction compared to vehicle-treated animals, yet this effect was statistically not significant (Figure 1c). Interestingly, on day 56 post CCI, Inco/A-treated rats showed a significant SFI recovery enhancement, as their walking pattern was statistically different to that of vehicle-treated animals (Figure 1c). Thus, repeated applications of Inco/A enhanced functional recovery after sciatic nerve damage, suggesting enhanced nerve regeneration.

Complementary assays are necessary to evaluate the effect of Inco/A on functional recovery and nerve regeneration post sciatic injury. Thus, we carried out Compound Muscle Action Potential (CMAP) recordings and histological analysis to examine the effect of Inco/A on our CCI model. CMAP recordings are frequently used to assess peripheral nerve regeneration [6,15,16]. Upon nerve injury, rats exhibited a substantial reduction in Action Potential (AP) amplitudes (approximately 80% reduction) in vehicle- and Inco/A-treated animals compared to the sham group, indicating impaired functionality of the motor units (Figure 2a). This effect was similar in all measures during the time course (days 3, 35, and 56 post CCI; Appendix A). A slight improvement trend, yet not statistically significant, was observed on day 56 post surgery in the Inco/A group compared to vehicle-treated animals (Figure 2a). A delay in AP latencies was also observed in animals post CCI in both the vehicle- and Inco/A-treated groups compared to sham animals, suggesting demyelination in the axons. This delay was observed in the three time point measures (Appendix A). Interestingly, on day 56 post surgery, AP latencies significantly decreased in the Inco/A-treated animals compared to the vehicle-injected rats. However, these latencies were still higher compared to those observed in the sham group. Thus, impaired but improved AP latencies were observed in animals treated with repeated administrations of Inco/A.

Shorter AP latencies obtained by CMAP in the Inco/A-treated group suggested enhanced axonal myelinization as an indicator of greater speed conduction. We then examined the myelinization ratio by performing toluidine blue staining in transverse sections of the sciatic nerve, close to the nerve injury (Figure 2c). By measuring the ratio between the axon and fiber diameter, it is possible to calculate the g-ratio, which can be correlated with the fibers’ maturation and their conduction velocity [17,18]. At the end point of our experiments, vehicle-treated animals showed an increased g-ratio compared to the sham group, indicating fiber demyelination as a consequence of nerve injury (Figure 2c,d). In contrast, Inco/A-treated animals showed an improved g-ratio, statistically significant and lower than that of the vehicle group (Figure 2c,d). These results correlate to the functional and electrophysiological assays (Figure 1 and Figure 2).

### 2.2. Inco/A Increased Nerve Myelinization and Diminished CGRP Expression

Previously, similar to our toluidine blue staining results, it was shown that BoNT/A increases myelinated fiber numbers in rats that underwent nerve injury [6,8]. As a complementary assay, we then performed immunolabeling of sciatic nerves against the 200 kD neurofilament protein (NF200) to evaluate whether multiple applications of Inco/A increase axonal regeneration after nerve injury. Figure 3 shows that CCI caused a substantial reduction (~2-fold signal reduction) in myelinated fibers in the sciatic nerve of vehicle-treated rats in comparison to the sham group. Moreover, the nerve fiber architecture was prominently different to that in the controls (Figure 3a). Inco/A-treated animals displayed myelinated fibers regeneration based on the NF200 signal, which was statistically different from that of the vehicle-treated animals (Figure 3a,b, Appendix A). However, nerve structure was not completely restored as a sequela of nerve injury, similarly to what is shown in Figure 2. Thus, all together, our data show that treating animals with two applications of Inco/A promotes regeneration of myelinated fibers after nerve injury in rats.

Apart from its effects on nerve regeneration, Inco/A inhibits the release of neuropeptides that are pain mediators such as substance P and calcitonin gene-related peptide (CGRP) [19,20]. We investigated whether CGRP was regulated after multiple administrations of Inco/A in rats. Following CCI, vehicle-treated animals showed a significant increase (approximately 3-fold) in CGRP expression compared to the sham group (Figure 3a,b, Appendix A). However, on day 56, after two administrations of Inco/A, sciatic nerves from the animals treated with the neurotoxin displayed a marked reduction in the CGRP signal compared to the vehicle-treated group (Figure 3a,b). These striking data suggest that treatment with Inco/A in the lesion area reduces the expression of the nociceptive neurotransmitter CGRP.

### 2.3. Inco/A Reduced Pro-Inflammatory Responses in Sciatic Nerves After CCI

As observed in Figure 3, nerve injury led to an increase in CGRP, which contributes to nociceptor sensitization. In fact, nociceptor sensitization occurs, in part, as a result of neuronal CGRP release which leads to leukocyte recruitment and therefore neuroinflammation [21,22,23]. It has been suggested that the inhibition of CGRP by BoNT/A and its regenerative effects can be explained in part by the diminishing inflammatory response during the regeneration process [6,11]. Thus, we examined whether Inco/A regulates immune responses in the sciatic nerve after nerve injury. Figure 4 shows that nerve injury led to a 2-fold increase in inflammatory cells at the site of injury in vehicle-treated rats in comparison to the sham group. Interestingly, perineural application of Inco/A exhibited a significant decrease (around 35%) in the inflammatory cell number compared to vehicle-treated animals (Figure 4b, Appendix A). Thus, Inco/A mediated anti-inflammatory responses which might contribute to nerve regeneration.

### 2.4. Inco/A Accelerated Schwann Cell (SC) Responses During Nerve Regeneration

The ability of the peripheral nervous system to regenerate nerves relies on the plasticity of SCs to reprogram into repair cells and proliferate, creating an optimal environment for axon regeneration [5,6,11]. We stained sciatic nerves from sham and vehicle- and Inco/A-treated rats to investigate whether in our model applications of Inco/A enhance Schwann cell proliferation when applied perineurally. Vehicle-treated animals showed a robust increase (10%) in S100β (S100 calcium-binding protein B) immunoreactivity compared to the sham group, suggesting the activation of the molecular and cellular machinery to recruit SCs into the injury site (Figure 5, Appendix A). Perineural application of Inco/A at the sciatic nerve displayed a significant reduction in the S100β signal on day 56 post CCI compared to the vehicle-treated animals (Figure 5, Appendix A). Thus, perineural and double application of Inco/A reverses SC migration/proliferation in the sciatic nerve after nerve injury based on the pro-repair marker S100β. Altogether, our findings showed that administering two doses of Inco/A perineurally to rats resulted in improved nerve regeneration, as measured by physiological and histological tests.

## 3. Discussion

Here, we have shown that Inco/A applied perineurally, to the sciatic nerve, accelerates functional recovery after nerve injury in rats by enhancing nerve regeneration. By applying Inco/A on site on the day of nerve injury followed by a second dose 21 days post-surgery, it was observed that rats showed an increased SFI and CMAP, indicating functional recovery from peripheral nerve injury. In contrast to vehicle-treated animals, rats treated with Inco/A also showed enhanced nerve myelination and restoration of nerve structure, the latter observed by using different histological approaches. Remarkably, injured sciatic nerves in the vehicle-treated group exhibited a significant pro-inflammatory response, which was partially mitigated in animals treated with Inco/A. This suggests that inhibiting the inflammatory response accelerates nerve regeneration, as previously speculated [11]. The reduced levels of CGRP observed in Inco/A-treated animals may contribute to this effect, as lower CGRP is associated with decreased inflammatory cell infiltration and a more favorable environment for nerve repair. Additionally, BoNT/A inhibits CGRP release from neurons by disrupting the SNARE-mediated exocytotic machinery, further supporting its anti-inflammatory role. Proliferation and migration of SCs are a key factor in nerve regeneration and are CGRP-dependent [5,6,24,25]. In contrast to previous studies, animals treated with Inco/A showed lower levels of immunolabeling against S100β in sciatic nerves compared to vehicle-treated rats. In fact, our finding suggests that, at this time point, two applications of Inco/A reverse the pro-regenerative repair phenotype of SCs, indicating virtually complete nerve regeneration. Moreover, a reduction in the S100β signal in sciatic nerves in Inco/A-treated animals can be explained by the low expression of CGRP at the injury site, which ultimately contributes to Schwann cell proliferation [25]. Our data showed that by applying two perineural injections of Inco/A on site after nerve injury, an enhancement in functional recovery and nerve regeneration can be achieved.

Functional recovery from peripheral nerve injury is typically poor in comparison to other tissues and limited depending on the site and the degree of nerve damage [11,26,27]. In fact, in most cases, nerve injuries persist as neuropathic pain, sensory loss, and incomplete functional recovery. Therefore, there is an urgent need to understand the mechanisms that contribute to nerve regeneration but most importantly to develop treatments that help patients to recover from functional disability. The use of BoNT/A has emerged as a promising therapy option for nerve regeneration based on pre-clinical studies, which makes it an attractive candidate not only to treat functional disability but also for other indications where nerve injury is involved such as peripheral neuropathic pain. However, the mode of action of BoNT/A in reducing pain and enhancing functional recovery is poorly understood. Recent clinical studies have shown evidence that supports the efficacy of BoNT/A in reducing mechanical allodynia [1,28] and enhancing functional recovery via nerve regeneration in animal models [5,6,7]. In particular, it has been demonstrated that SCs play an important role in the nerve regeneration process accelerated by BoNT/A, and it has been proposed that pro-inflammatory responses play an important role during nerve damage and recovery [5,6,11,24]. However, a full understanding of the mode of action in both pre-clinical and clinical studies is the focus of current investigations.

Most of the pre-clinical studies investigating functional recovery and nerve regeneration were carried out by administering a protein complex that includes BoNT/A, and only a few studies have characterized the therapeutic potential of using the purified version of BoNT/A, such as Inco/A [6,29,30]. In this study, by using a BoNT/A formulation devoid of complexing proteins, which can bind to non-neuronal cells [31], we ensure that the observed biological effects are attributable to the neurotoxin itself. Moreover, pre-clinal studies differ regarding site, time, and dose of administration of BoNT/A. While in some studies BoNT/A was injected at the periphery (such as intradermal, subcutaneous, perineural, and interneural), other studies investigated treatment effects after central nervous system (spinal) application [5,6,8,32,33]. Therefore, besides common observations indicating the positive effect of BoNT/A on nerve regeneration, it has been challenging to elucidate in detail the mode of action of BoNT/A on nerve regeneration, particularly in its purified form (Inco/A).

Marinelli et al. (2010) and Cobianchi et al. (2017) were the first to report functional recovery and axonal regeneration using BoNT/A after nerve injury [5,6] (Appendix A). When BoNT/A was injected intraplantarly into the hind paw of mice 5 days post CCI, Marinelli et al. (2010) observed that functional recovery (by measuring the Sciatic Static Index, SSI) was improved already on day five post injection compared to the saline-solution-injected group [5]. This effect was maintained until day 81 post CCI, where they observed an accelerated functional recovery of up to 40% in absolute numbers (−70 and −30 SSI values on days 10 and 80 post CCI, respectively) [5]. Similarly, other studies injecting BoNT/A intraneurally showed faster recovery [8,10]. The improved functional recovery effect is surprisingly faster compared to the effect we observed in rats where the effect of Inco/A was already visible, yet statistically insignificant, on day 35 post CCI following two on-site administrations of the neurotoxin. However, it is important to note that in their study, the SSI values in the vehicle control group changed slightly compared to the first measure where animals reached an SSI value of −80 (the SSI value on days 40 and 60 days post CCI was approximately −70), while in our model, vehicle-injected rats showed an SFI value of approximately −50 on days 36 and 56 post nerve injury. Furthermore, the functional recovery measures and analysis differ; while we used the SFI, they studied the SSI, where the animal is in a static position, reducing the effect of gait velocity [14]. Thus, discrepancies could be the result of different administration modes and functional recovery assays, and there may be species-specific factors that enhance intrinsic nerve recovery in rats compared to mice even in the absence of any treatment.

Similar to previous studies, we observed that administration of Inco/A perineurally significantly increases the NF200 signal in the sciatic nerve after nerve injury [6,32]. Even though our nerve injury models differ (nerve crush vs. CCI), the remarkable increase in myelinated fiber regeneration in the presence of BoNT/A indicated its powerful effect. Cobianchi et al. used a purified version of BoNT/A, similar to our study [6]; therefore, the possible effect of accessory proteins was discarded [31]. We observed that the increase in the NF200 signal was stable and maintained even on day 56 post surgery, as previously reported [8]. Moreover, the positive effect of Inco/A on accelerating nerve myelinization was confirmed by measuring the g-ratio using toluidine blue. It remains to be seen whether this sustained signal in myelinated fibers occurred as an effect of the first dosing or whether it was enhanced by the application of the second dose of Inco/A on day 21 post-surgery. However, our CMAP measures indicate that shorter latencies are achieved only on day 56 post nerve injury, suggesting that the second dose may have boosted nerve regeneration. In the Cobianchi et al. study, faster AP responses were observed in the crushed group, but values did not reach significance [6]. However, in their CMAP measurements, they noticed that mice subjected to CCI showed a significant increase in AP amplitude 3 weeks after CCI and 16 days post intraplantar BoNT/A administration [6]. Similar results were observed when BoNT/A was injected intraneurally [8]. This raises the question of whether intraplantar and intraneural administrations of the toxin are more efficient than perineural application of BoNT/A. The impact of local administration of Inco/A on nerve regeneration and sprouting at the periphery remains to be studied, especially considering that nerve trauma leads to collateral sprouting, where adjacent undamaged nerve fibers, particularly C-fibers, invade denervated areas [34]. Moreover, histological data evaluating nerve regeneration at the sciatic nerve after nerve trauma in animal models treated with BoNT/A at the periphery would be needed to rule out alternative mechanisms by which animals recover AP amplitude. Notably, as we did not observe complete nerve structure regeneration despite higher myelinization, this suggests that the changes in CMAP amplitudes are due to lack of regeneration of other types of fibers, including C-fibers.

CGRP levels were considerably downregulated in the sciatic nerve after Inco/A administrations in our CCI model. These results differ from the study of Cobianchi et al., where changes in the CGRP expression were not observed after nerve crush [6]. A reduction in the nociceptive neuropeptide CGRP explains in part the accelerated functional recovery results, where animals not only display nerve regeneration but also a readout of pain reduction, as previously reported in animal models. It would be interesting to investigate the dynamic expression of CGRP after nerve injury, as well as to study which fibers contribute most to CGRP release, as both C-fibers and Aδ-fibers and a subpopulation of high-threshold mechanosensitive Aβ-fibers express this neuropeptide [23].

A reduction in CGRP levels in the presence of Inco/A might shorten the inflammatory phase after nerve injury, allowing nerve repair processes to proceed more efficiently. Nociceptor-derived CGRP can modulate leukocyte dynamics during tissue healing (regeneration) after injury [35]. If CGRP contributes to the increase in leukocyte migration into the injury site, immune responses can lead into an enhanced release of pro-inflammatory cytokines and other inflammatory agents (by, e.g., macrophages and monocytes) that contribute to allodynia and hyperalgesia of intact sensory neurons after nerve injury [21], progressive neuronal death [32], and possibly limiting the nerve generation process, as previously proposed [6,11]. In fact, we observed that leukocyte migration on day 56 post CCI into the site injury was tremendously decreased after treating animals with Inco/A. To our knowledge, this study is the first to demonstrate a reduction in leukocyte infiltration in the presence of Inco/A at the injury site, besides previous studies that had proposed the anti-inflammatory effects of BoNT/A [12,36]. Leukocyte infiltration into the sciatic nerve following injury has been well characterized, identifying the presence of neutrophils, macrophages, and T lymphocytes using specific markers [37]. Neutrophils are the first immune cells recruited during the acute phase of nerve injury, partially due to the chemoattractant properties of SP and CGRP released by sensory neurons [38]. Neutrophils also trigger the recruitment of macrophages and T cells, which are important for later stages [38]. In the present study, we did not analyze leukocyte infiltration in the early hours or days following nerve injury; therefore, we cannot conclude whether Inco/A plays a role in neutrophil recruitment. It would be interesting to investigate whether Inco/A contributes to the acute phase by recruiting neutrophils or to later stages involving macrophages and T lymphocytes. In fact, macrophages play a critical role in clearing myelin and cellular debris, and their infiltration is regulated by neutrophils, SCs, and sensory neurons [38,39,40]. These cells exhibit phenotypic plasticity, adopting either a pro-inflammatory (M1) or anti-inflammatory (M2) profile [37,38]. While macrophages contribute to tissue repair and homeostasis, they also contribute to peripheral neuroinflammation, resulting in neuropathic pain [38,41].

As shown in Figure 4, animals treated with two administrations of Inco/A displayed a reduced number of inflammatory cells in the sciatic nerves after injury. One hypothesis is that the reduction may be attributed to a decrease in pro-inflammatory M1 macrophage infiltration due to their role in neuroinflammatory conditions [42]. T lymphocytes contribute to inflammatory responses at the latest stage of nerve injury [43]. Similar to macrophages, T lymphocytes can produce pro-inflammatory (Type 1 helper, Th1 cells) or anti-inflammatory (Type 2 helper, Th2 cells) responses [38,43]. Thus, another hypothesis is that Inco/A modulates T-cell infiltration after nerve injury, particularly by decreasing the Th1 cell population. At this stage, we can only speculate on the specific lymphocyte subpopulations regulated by Inco/A. Further studies using specific markers for neutrophils, macrophages, and T lymphocytes are necessary to confirm which leukocyte populations are regulated by Inco/A, at which stage [44], and whether multiple administrations influence the inflammatory responses in peripheral nerve injury models.

Schwann cells are the glial support cells that myelinate peripheral axons and post nerve injury transform into a repair phenotype that guides axon regeneration. Previous studies have reported increased expression of Schwann cell activation markers such as S100β after nerve injury when treating animal models with BoNT/A, indicating enhanced SC proliferation and dedifferentiation into nerve-repair-mode cells [5,6,8,10,32]. In contrast, we observed that after two perineural applications of Inco/A, the S100β marker was decreased on day 56 post CCI (Figure 5) to a level which was similar to the sham. This surprising result could indicate that nerve regeneration in our model was not only enhanced but probably completed. We speculate that at this end point, the proliferation and dedifferentiation of SCs were not needed any longer. This effect can also be explained by the reduced number of leukocytes in the injury region, potentially resulting in a lower concentration of inflammatory mediators responsible for promoting SC recruitment [45,46]. Nevertheless, as discussed above, given the lack of precise characterization of the immune cell population influenced by Inco/A across various time points, we are limited to speculation regarding the underlying mechanism. In previous studies, where the increase in S100β was sustained even after 56 days post injury, the authors did not study the immune response at the site of nerve injury [8,10]. Moreover, differently to our study, a non-purified version of BoNT/A was used. The use of various types of BoNT/A can result in differing interpretations, particularly due to the potential added effect of protein complexes in non-purified forms which could lead to inflammatory responses [31]. Finally, the BoNT/A administration was different (intraneural vs. perineural). Thus, it remains to be studied whether differences are attributable to the use of different types of BoNT/A or different experimental models or if a second administration of Inco/A is needed to reverse the effects of SC proliferation and dedifferentiation.

One limitation of our study is that the experiments were performed only in male rats. There is emerging evidence indicating that the development of strategies to treat conditions such as pain should be studied in both sexes [47,48,49,50]. Thus, further studies are needed to determine whether functional recovery and nerve regeneration are equal in both sexes under our experimental conditions. Future research should incorporate both males and females to provide a more comprehensive understanding of the effects of Inco/A on nerve regeneration. Given the significant potential of BoNT/A for treating nerve injury conditions, such as peripheral neuropathic pain [51], it is essential to translate the findings from rodent studies involving both sexes to design more effective clinical trials. This approach will help address sex differences in pain and regeneration research, ultimately enhancing the clinical applicability of BoNT/A treatments, including Inco/A. Moreover, our histological studies were conducted only at the end of our experimental model (56 days post CCI); therefore, more data from intermediate time points would be informative to study the dynamic expression of the different studied markers and when the changes occur, therefore, more data from intermediate time points would be informative to study the dynamic expression of the different studied markers and when the changes occur, particularly indicators of leukocytes or of SC migration. As discussed above, it would be of interest to study the mechanisms that occur at the periphery, where induced (mechanical) pain and nerve reinnervation in the skin take place [34,52]. Histological studies investigating neuronal, inflammatory, and glial markers could be carried out to study the effect of Inco/A on nerve regeneration at the periphery. We demonstrated that two doses of Inco/A virtually recovered nerve regeneration based on the pro-repair S100β marker in SCs and the SFI. Although rodents and humans share similar molecular and cellular mechanisms (including SC responses after nerve injury), important differences in regeneration rates and anatomical structures need to be considered carefully when translating rodent studies to clinical applications in humans [53,54].

Our findings may hold clinical relevance, particularly regarding the treatment of nerve injuries and chronic neuropathies. The success of nerve regeneration depends on the severity of the peripheral nerve injury and the timing of treatment. Administering BoNT/A locally during the acute phase of injury (or even prophylactically, such as prior to a planned surgery) can prevent deficits in functional recovery and enhance nerve regeneration [8]. Additionally, subsequent administrations of BoNT/A could further improve recovery after nerve trauma [11], potentially by reducing pro-inflammatory responses in the non-acute phase. Innovative treatments that promote axonal recovery are particularly attractive for patients with acute nerve injury, including post-surgical damage [8]. The relevance of our study might extend to post-surgical settings, where the perineural administration of BoNT/A could be adapted to mitigate nerve damage and accelerate recovery following surgical procedures. In a recent publication, it was discussed that there is a medical need to find alternative treatments for patients with chronic neuropathies such as diabetic neuropathic pain (DPN) [55]. Patients with such debilitating conditions can display limited affected areas where BoNT/A can be administered subcutaneously or intradermally, and therefore, safe and effective alternative delivery methods, such as perineural administration, are required [55]. In fact, our research employs a similar approach to an ongoing phase II clinical trial involving DPN patients [55], supporting the potential clinical adoption of perineural BoNT/A treatment which could address an unmet medical need. Thus, by administering BoNT/A (Inco/A in this study) at the site of nerve injury (perineurally) during the acute phase, followed by a second dose in the non-acute phase, we not only demonstrate that BoNT/A is a promising candidate for treating functional disabilities resulting from nerve injury but also support its potential as a therapeutic agent for other nerve-injury-related conditions, such as peripheral neuropathic pain.

## 4. Materials and Methods

### 4.1. Animals

All animal study protocols were reviewed and approved by the corresponding authorities (Animal Care and Use Committee of HD Biosciences, Shanghai, China). Male SD rats (200–230 g) were obtained from Vital River Laboratory Animal Technology Co., Ltd. (Charles River) (Beijing, China). Rats were kept in cages (4–5 animals per cage, 42 × 31 × 18 cm) with bedding under controlled temperature (20–25 °C) and humidity (40–70%) conditions and a standard 12/12 h light–dark cycle. All animals were provided with free access to purified water and standard certified rodent chow (Beijing Keao Xieli Feed Co., Ltd.) (Beijing, China). The appearance and activity of animals were carefully monitored daily by a veterinarian. Unless specified, all animals underwent all assays described, and all readout values from these assays were included in the analysis.

### 4.2. Chronic Constriction Injury (CCI) Surgery of the Sciatic Nerve

With small modifications, we adapted the CCI model which was developed by Bennett and Xie (1988) [56]. Before surgery, rats were anesthetized by inhalation of isoflurane. The right sciatic nerve was exposed in the gluteal region and loosely ligated with four 5-0 silk sutures, spaced 1.0 mm apart and positioned 1–1.5 mm upstream of the nerve trifurcation into the tibial, sural, and common peroneal nerves. A 20 µL solution of saline or Inco/A was applied at the ligation site shortly after the sutures were placed (on-site application; 4U of Inco/A). The wounds were then closed in layers. After recovering from anesthesia, the animals were returned to their individual cages. In the sham group, surgery followed the same procedure, except the nerve was not ligated. All procedures were performed aseptically.

### 4.3. Drugs

IncobotulinumtoxinA (Inco/A; Xeomin^®^, Merz Therapeutics GmbH, Frankfurt am Main, Germany) was diluted to a final concentration of 0.2 U/µL in saline solution. Inco/A (20 µL) was placed at the site of ligation (on-site application) on day 0 (day of surgery) and day 21 post CCI.

### 4.4. Behavioral Tests

Animals were randomized into three groups of 12 rats based on their body weight at the beginning of the study. Experimenters were blinded to the treatment groups during data collection and analysis. Functional readouts (Sciatic Functional Index and Compound Muscle Action Potential, see below) were performed on days 3, 35, and 56 post CCI. Tissue collection was carried out on day 56 after surgery for histological assessments (see below).

### 4.5. Sciatic Functional Index (SFI)

The SFI test was conducted in a confined corridor (100 × 10 × 20 cm) with a dark box at one end. White paper was placed on the floor of the corridor. Prior to surgery, all rats were trained to walk through the corridor. Once the animals learned to walk along the runway without stopping, their footprints were recorded. The hind paws of the rats were pressed onto a sponge soaked with finger paint. Animals left their hind footprints on the paper when walking through the corridor. SFI variables were measured: paw length (PL), the distance from heel to toe; toe spread (TS), the distance between the first and fifth toes; and intermediary toe spread (ITS), the distance between the second and fourth toes. All measurements were taken for both the experimental (E) and control (N) paws. The SFI value was then calculated using the formula SFI = −38.3[(EPL − NPL)/NPL] + 109.5[(ETS − NTS)/NTS] + 13.3[(EIT − NIT)/NIT] − 8.8, where EPL is the experimental paw length, NPL is the normal paw length, ETS is the experimental toe spread, NTS is the normal toe spread, EIT is the experimental intermediary toe spread, and NIT is the normal intermediary toe spread. An SFI value of 0 indicates normal function and −100 indicates total impairment. An index score of −100 was assigned when no footprints were measurable. For each walking track, three footprints were analyzed by a single observer, and the average of the measurements was used in the SFI calculations.

### 4.6. Compound Muscle Action Potential (CMAP)

CMAP recordings were performed on all animals after anesthesia was administered via intraperitoneal injection of Zoletil-50 (40 mg/kg) combined with Xylazine (10 mg/kg). CMAPs were recorded in the gastrocnemius muscle using surface stimulation through the tendon–belly method. The sciatic nerve was stimulated with bipolar electrodes placed on the skin (premoistened with gel), between the ischial tuberosity and the major trochanter, parallel to the sciatic nerve. The active and reference monopolar needle electrodes were inserted into the mid-belly and the muscle tendon surface, respectively. A ground electrode was clamped to the skin between the stimulating and recording electrodes. Stimulations lasting 0.02 ms were applied at increasing intensities until a maximal CMAP response was achieved. The recording was repeated three times, and the amplitude and latency of the CMAP were averaged for each rat. Data acquisition and analysis were performed using PowerLab (ADI, PL3516) (ADInstruments, Dunedin, New Zealand). The Nerve Conduction Velocity Module (ADI, FE180 and FE232) used with the PowerLab system was used to measure nerve conduction velocity (ADInstruments, Dunedin, New Zealand).

### 4.7. Tissue Collection

On day 56 after CCI, animals were euthanized by CO_2_. Animals were perfused with cold saline, and sciatic nerves were collected and fixed in 4% paraformaldehyde. Tissue collection included ligatures (0.5 cm from both sides of external ligatures).

### 4.8. Hematoxylin and Eosin Staining (H&E Staining)

Sciatic nerves were collected as described above. Longitudinal sections 20 µm thick were acquired. Sections were stained with Hematoxylin (BASO# BA4041, BaSO Biotech, Taipei, Taiwan) for 90 s. Sections were rinsed with distilled water to remove the excess. Then, sections were stained using Eosin staining solution (BASO# BA4022, BaSO Biotech, Taipei, Taiwan) for 30 s and were washed with distilled water for 30 s. Sections were dehydrated, and mounting was performed for subsequent image analysis. H&E-stained images were scanned using a Leica Aperio GT450 (objective 20×) (Leica Biosystems, Deer Park, TX, USA). Images were analyzed using HALO AI (version 3.3.2541, Indica labs, Albuquerque, NM, USA). Cell counting was carried out by using the “Nuclei Seg” module. The Nuclei Phenotyper tool was trained to identify leukocytes, and the number of inflammatory cells was counted automatically.

### 4.9. Toluidine Blue Staining

Sciatic nerves (about 0.5 cm) were put into fixative containing 4% paraformaldehyde and 0.5% glutaraldehyde in PBS. Nerve tissue was then incubated in Osmium tetroxide (OsO4) solution (Energy-Chemical, Shanghai, China) for 1 h. The tissue was then dehydrated by submersing in ethanol at increasing concentrations from 30% to 100%, followed by embedding in epoxy resin. Embedded tissue was incubated overnight at 60 °C. Then, 2 µm thick sections were obtained using a Leica RM2235 Microtome (Leica Biosystems, Deer Park, TX, USA). Sections were then mounted on glass slides (Fisherbrand Superfrost^TM^ Plus, Fisher Scientific, Pittsburg, PA, USA) and stained with toluidine blue (Sinopharm Beijing, China) for 5 min. Sections were gently dipped into a deionized water jar, and this was repeated 3–4 times to remove excess toluidine blue solution and until sections were clear. Slides were dried out before image acquisition (Leica DM2500) (Leica Biosystems, Deer Park, TX, USA). For each sample, three fields were randomly imaged under a 100× oil immersion objective. Images were analyzed using HALO AI (version 3.3.2541, Indica labs, Albuquerque, NM, USA). For morphometric analysis of sciatic nerves, “Axon v1.5” was used to measure myelinated fiber and axonal diameter values. The degree of myelination was calculated by obtaining the g-ratio (quotient axon diameter/fiber diameter; Figure 2), as in [17,18].

### 4.10. Immunofluorescence Staining and Image Analysis

Sciatic nerves were collected and placed in 4% paraformaldehyde (PFA) in phosphate-buffered saline (PBS, pH 7.4) overnight, followed by 3 × 10 min washing in PBS. Samples were then incubated in 30% sucrose in PBS overnight. Sciatic nerves (about 1 cm) were frozen in Optimal Cutting Temperature (OCT) solution, and longitudinal cryostat microtome sections (20 µm) were taken (Leica CM3050S Microtome, Leica Biosystems, Deer Park, TX, USA) and mounted directly on glass coverslips. Tissue sections were air-dried for 1 h at room temperature before being processed for immunofluorescence. Samples were blocked with Goat Serum (1%, Kangyuan #KY-01021) (Yaanda Biotechnology Co., Ltd., Beijing, China) and Triton (0.3%, Sigma-Aldrich, Burlington, VT, USA) in PBS for 1 h. Tissue sections were then incubated overnight at 4 °C with the following primary antibodies: anti-CGRP (Mouse, Abcam, ab81887; 1:100, Cambridge, UK), anti-NF200 (Rabbit, Cell Signaling Technology, 55453S; 1:200, Leiden, The Netherlands), and anti-S100β conjugated to Alexa Fluor^®^ 594 (Rabbit, Abcam, ab307851, Cambridge, UK). Sections were then washed 3 × 10 min with 0.3% in PBS followed by 1 h incubation at room temperature with the following secondary antibodies: Goat anti-Rabbit (IgG) 594 Alexa Fluor (Abcam, ab150080; 1:500, Cambridge, UK), Goat anti-Mouse (IgG) 488 Alexa Fluor (Abcam, ab150113; 1:500, Cambridge, UK), and DAPI nuclear stain (Abcam, ab104139; ready to use, Cambridge, UK). Samples were washed 3 × 10 min in wash buffer (EnVision FLEX, #K8007, Agilent Dako, Santa Clara, CA, USA) Images were taken on a ZEISS Axioscan7 (Zeiss, Oberkochen, Germany) scanning microscope using a 20× objective (0.17 µm/pixel) using the following wavelengths: 420 laser, 460/550 nm emission/excitation; 594 laser, 618/590 nm emission/excitation; and 488 laser, 520/488 nm emission/excitation). For image analysis, HALO AI software was used (version 3.3.2541; Indica Labs, Inc., Albuquerque, NM, USA). Image regions were annotated using HALO annotation tools, and the area of nonspecific staining was excluded using the scissor tool. We next used Area Quantification FL (Version 2.3.3) for the detection of positive areas with different fluorescence markers

### 4.11. Statistical Analyses

All data was plotted as the mean ± standard error of the mean (SEM). Data analysis was performed using GraphPad Prism (Version 10.4.1); data sets were tested for normality using the D’Agostino–Pearson omnibus normality test, and next, equal variance was verified using the Brown–Forsythe test. Depending on the outcome of the normality test, either a One-way ANOVA or a Kruskal–Wallis test was performed. Vehicle groups were used as the control in all multiple comparison analyses, as the primary objective of the study was to assess the effects of Inco/A treatment relative to the vehicle group after nerve injury. Appropriated post hoc tests (Holm-Šídák’s and Dunn’s tests following One-way ANOVA and Kruskal-Wallis, respectively) were then applied to identify significant differences between groups.

## 5. Conclusions

We administered two injections of Inco/A, one on the day of the nerve injury followed by a second dose 21 days post CCI, and evaluated its effect on nerve regeneration and functional recovery. Surprisingly, we observed that two injections of Inco/A, apart from restoring functional recovery as previously shown, reduced pro-inflammatory responses by measuring the number of leukocytes in the injured sciatic nerve. Moreover, in contrast to other studies, we observed that S100β levels, a marker of Schwann cell dedifferentiation and proliferation, were similar to control levels, suggesting that nerve regeneration was virtually restored. Our data show for the first time that two applications of Inco/A decrease leukocyte infiltration at the site of injury, indicating that apart from the role of Schwann cells in nerve regeneration, anti-inflammatory responses mediated by leukocyte migration are crucial during nerve regeneration and functional recovery. This study supports the current picture that proposes the use of BoNT/A as a safe and efficient treatment to improve recovery in patients with peripheral nerve injury.

## Figures and Tables

**Figure 1 ijms-26-07482-f001:**
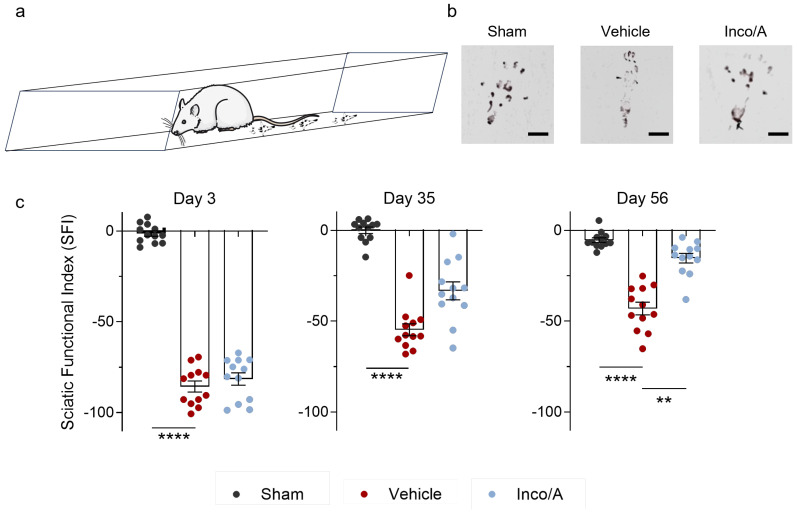
Inco/A enhanced functional recovery in rats after chronic constriction injury (CCI). (**a**) Cartoon representing the Sciatic Functional Index (SFI) assay. Animals were placed in a confined corridor, and hind footprints were recorded and analyzed. The cartoon figure was produced using Servier Medical Art (http://smart.servier.com/). (**b**) Representative examples of footprints (ipsilateral paw) on day 56 post CCI (scale bar: 1 cm). (**c**) Histograms showing the functional recovery of rats treated with Inco/A after CCI measured as SFI. Note that after day 3 post CCI, animals showed an increased SFI absolute value, indicating neuropathic pain and functional disability induced by CCI. On day 56 after CCI, rats showed decreased SFI values compared to animals treated with the vehicle, suggesting increased functional recovery. (Day 3 post CCI: One-way ANOVA; **** *p* < 0.0001. Days 35 and 56 post CCI: Kruskal–Wallis test; ** *p* = 0.006, **** *p* < 0.0001.) Each dot represents an individual animal (*n* = 12 for each group). Data are presented as the mean ± standard error of the mean.

**Figure 2 ijms-26-07482-f002:**
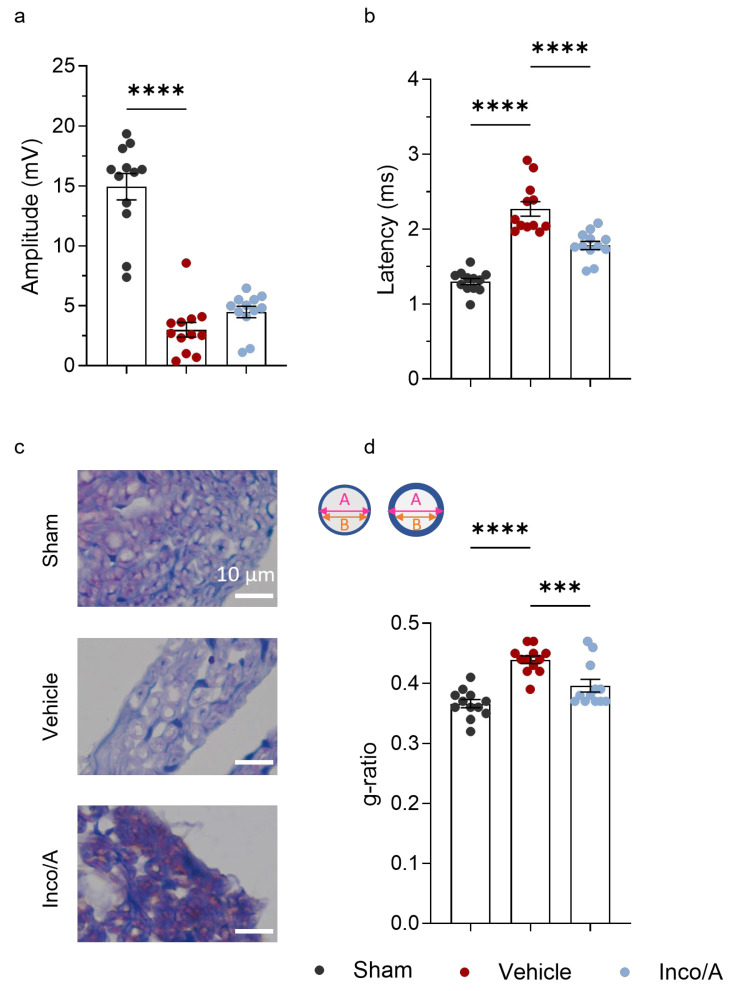
Inco/A enhanced sciatic nerve regeneration and nerve myelinization in rats after chronic constriction injury (CCI). (**a**) Histograms showing amplitude of Compound Muscle Action Potential (CMAP) recordings on day 56 post CCI. Note that CCI decreases amplitude values, suggesting nerve damage (Kruskal–Wallis test; **** *p* < 0.0001). (**b**) Column bars showing that Action Potential latencies are increased after nerve injury in rats. Animals treated with Inco/A displayed shorter latencies, indicating nerve regeneration (One-way ANOVA; **** *p* < 0.0001). (**c**) Representative images of transverse sections of the sciatic nerve using toluidine blue staining in sham and vehicle- and Inco/A-treated rats. Scale bar: 10 µm. (**d**) g-ratio analysis indicated higher myelination on day 56 post CCI in the Inco/A group compared with vehicle-treated animals (One-way ANOVA; *** *p* = 0.0007, **** *p* < 0.0001). Insert represents a slightly (**left**) and a highly (**right**) myelinated fiber, where *A* indicates the quotient axon diameter and *B* the fiber diameter. Each dot represents the value from an individual animal (*n* = 12 in each group studied). Data are presented as the mean ± standard error of the mean.

**Figure 3 ijms-26-07482-f003:**
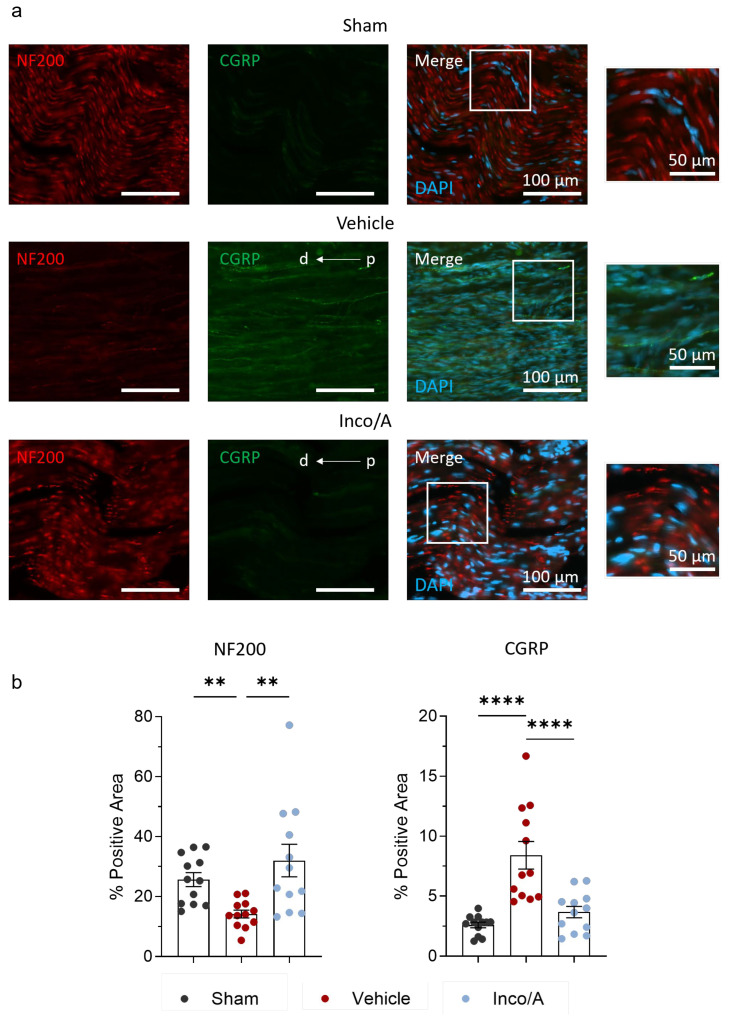
Inco/A restored regeneration of myelinated fibers and decreased calcitonin gene-related peptide (CGRP) in injured sciatic nerves in rats. (**a**) Representative immunostainings showing the 200 kD neurofilament protein (NF200) and CGRP expression in longitudinal sections in sciatic nerves from sham and vehicle- and Inco/A-treated animals 56 days post CCI (scale bar: 100 µm). The white squares highlight a section added as an insert to the right for all groups (scale bar of the inserts: 50 µm). In all images, the right side represents the proximal (p) area relative to the injury site, while the left side corresponds to the distal (d) area. (**b**) Histograms showing the quantification of NF200 (**left**) and CGRP (**right**) fluorescence signal in sciatic nerves from sham and vehicle- and Inco/A-treated animals (for NF200, Kruskal–Wallis test, ** *p* < 0.01; for CGRP, One-way ANOVA, **** *p* < 0.0001). Each dot represents the value of the percentage of positive area (see Section 4) in sections from an individual animal (*n* = 12 for each group). Data are presented as the mean ± standard error of the mean.

**Figure 4 ijms-26-07482-f004:**
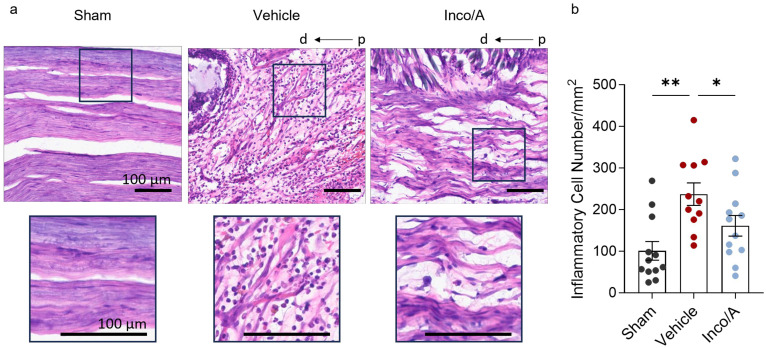
Inco/A reduced inflammatory responses in the injured sciatic nerve. (**a**) **Above**, representative images of longitudinal sections stained with Hematoxylin from sciatic nerves from sham and vehicle- and Inco/A-treated animals 56 days post CCI (scale bar: 100 µm). **Below**, a magnification of the images above (black square) is included as an insert for all groups (scale bar: 100 µm). Note the structural damage in the sciatic nerve following CCI. In all images, the right side represents the proximal (p) area relative to the injury site, while the left side corresponds to the distal (d) area. (**b**) Column bars displaying that the number of leukocytes (inflammatory cell number) in sciatic nerves is clearly increased post CCI in vehicle- versus Inco/A-treated animals (One-way ANOVA; * *p* = 0.03, ** *p* = 0.001; one outlier was discarded for the analysis according to Grubb’s outlier analysis). Each dot represents the value of an individual animal (*n* = 12 for each group, except in the vehicle group where 1 outlier was discarded). Data are presented as the mean ± standard error of the mean.

**Figure 5 ijms-26-07482-f005:**
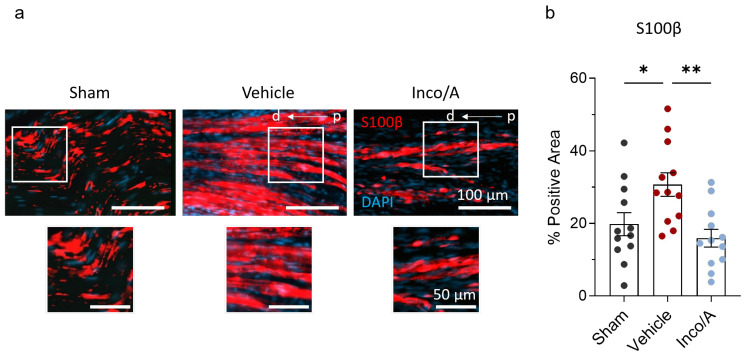
Inco/A reversed Schwann cell migration and proliferation into the sciatic nerve post CCI. (**a**) **Above**, representative immunostainings for S100β in sciatic nerve (longitudinal sections) from sham and nerve-injured rats in the absence or presence of Inco/A (scale bar: 100 µm). **Below**, a magnification of the images above (white square) is included as an insert (scale bar: 50 µm). In all images, the right side represents the proximal (p) area relative to the injury site, while the left side corresponds to the distal (d) area. (**b**) Histograms displaying the reduction in the S100β signal in Inco/A-treated animals compared to rats treated with vehicle. Note that S100β immunolabeling is similar between the sham and Inco/A-treated groups (One-way ANOVA; * *p* = 0.01, ** *p* = 0.002). Each dot represents the values from staining of an individual animal (*n* = 12 for each group). Data are presented as the mean ± standard error of the mean.

## Data Availability

Raw data are available on reasonable request from the corresponding author.

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
