# Peer review of "The Neuroregenerative Effects of IncobotulinumtoxinA (Inco/A) in a Nerve Lesion Model of the Rat"

_ijms, 2025, doi:10.3390/ijms26157482_

Round 1
Reviewer 1 Report (Previous Reviewer 2)
Comments and Suggestions for Authors
The revised manuscript is significantly improved and addresses most of the reviewer issues from the original peer review. The authors now present more robust evidence for the neuroregenerative and anti-inflammatory effects of IncobotulinumtoxinA (Inco/A) in a rat chronic constriction injury (CCI) model. Their inclusion of additional, more detailed histological evaluation, quantitation of immune cell infiltration, and longer discussion of Schwann cell activity offers a more complete mechanistic viewpoint. In addition, the data presentation and figure clarity are considerably improved and better annotated.
1. While the study demonstrates improved regeneration at day 56, the addition of intermediate time points (day 14 or 35) for markers such as NF200, CGRP, or S100β would be helpful to determine the time course of Inco/A's effects. This would determine whether the observed benefits are due to delayed or sustained effects.
2. The paper concentrates on histology of the nerve trunk but does not evaluate reinnervation of the distal target tissues (i.e., muscle or skin). The addition of data regarding target reinnervation or functional sensory return would enhance translational impact.
3. With the noted downregulation of CGRP and leukocyte infiltration, addition of behavioral pain assays (e.g., von Frey, thermal withdrawal) would validate the assertion of Inco/A's analgesic effects. This would be consistent with prior BoNT/A studies and enhance conclusions regarding functional outcomes.
4. Although the text explains suitable statistical tests, not all of the figures state clearly which groups are being compared, or include post-hoc markings (e.g., asterisks with brackets). Including these would make figures clearer and easier to interpret visually.
5. The authors talk about NF200 (myelinated fibers), but since CGRP is mainly expressed in small-diameter sensory neurons (C-fibers and Aδ fibers), it would be valuable to include or provide data on these populations. Double staining for peripherin or IB4 could more forcefully support conclusions regarding sensory axon regeneration.
Comments on the Quality of English LanguageThe manuscript is quite well-written and clear, but there are a number of places where language polishing would help to enhance clarity and professional tone.
Author Response
Please see the attachment.

Reviewer 2 Report (New Reviewer)
Comments and Suggestions for Authors
The authors have demonstrated that IncoA helped neuroregeneration after CCI, as reflected in improvements in multiple aspects such as SFI and CMAP, and reduction in inflammtory responses. Whilst this study is of interest and potential clinical utility, some concerns need to be addressed in the revision of this MS.
Major:
- A control (IncoA injection in sham animals not inflicted with CCI) is lacking. I ask this question because IncoA itself could change a number of parameters. Of course I would not expect it would cause a positive value in SFI, but it is possible that it would affect expression of a number of inflammtory molecules. Thus, authors either need to show that IncoA does not affect these parameters or discuss that it is highly unlikely to do so according to previous literature.
- Why use IncoA instead of BotoxA may have to be emphasized more and in the Discussion, explain more about the putative molecular mechanism of IncoA in neuroregeneration.
Minor:
- Past tense to be used througout the text when authors' own results are descirbed.
- page 233, should be "restoration"
Round 2
Reviewer 1 Report (Previous Reviewer 2)
Comments and Suggestions for Authors
This revised manuscript is a considerable enhancement from the original submission. The authors have successfully dealt with all major reviewer concerns from the first round of peer review. They have elaborated on their mechanistic discussion, made their experimental rationale clearer, and enhanced figure quality and data presentation.
Minor Comments
1. The manuscript switches back and forth between using "BoNT/A" and "Inco/A" in referring to the experimental agent. Although it is made clear early on that Inco/A is a purified BoNT/A, it would enhance clarity to refer to it consistently as "Inco/A" throughout the manuscript when talking about the experimental agent.
2. The authors take decreased S100β expression at Day 56 to indicate full nerve regeneration. Although this is possible, it is somewhat speculative. Granting that decreased Schwann cell marker expression can also be representative of other cell dynamics would be a more balanced interpretation.
3. A few graphs have proximal/distal labeling in the legend, but not always with clear visual distinction in the figures themselves. The addition of scale bars to main panels (not inserts) and in-figure annotations (e.g., arrows or "proximal/distal" labels) would increase figure clarity and readability for readers.
Author Response
Please see the attachment.

This manuscript is a resubmission of an earlier submission. The following is a list of the peer review reports and author responses from that submission.
Round 1
Reviewer 1 Report
Comments and Suggestions for Authors
As a side note…I commend the authors for the study and for its thoughtfulness. Toxin use clinically is expanding into a board spectrum of disorders and it isn’t intuitive that this ( at such an early investigative stage ) may have other physiological impacts open its use in other disorders ( peripheral nerve injuries etc). Solid basic science
Author Response
Comments 1:
As a side note…I commend the authors for the study and for its thoughtfulness. Toxin use clinically is expanding into a board spectrum of disorders and it isn’t intuitive that this ( at such an early investigative stage ) may have other physiological impacts open its use in other disorders ( peripheral nerve injuries etc). Solid basic science
Response 1:
Thank you very much for your insightful comment, we really appreciate it. We truly believe that botulinum toxin use in clinical practice does indeed hold vast potential across a spectrum of disorders, and we are enthused by the possibilities it seems to present for conditions such as peripheral nerve injuries. Your acknowledgment of our work is highly encouraging and motivates us to continue exploring these promising avenues.
Reviewer 2 Report
Comments and Suggestions for Authors
This manuscript presents a well-designed preclinical study investigating the neuroregenerative and anti-inflammatory effects of IncobotulinumtoxinA (Inco/A) in a rat chronic constriction injury model. The use of multiple assessment methods, behavioral, electrophysiological, and histological, strengthens the findings, which suggest that repeated Inco/A administration improves functional recovery and reduces inflammation at the injury site.
The data are generally convincing, and the writing is clear. However, key issues need to be addressed before publication. These include better justification of the dosing strategy, further interpretation of S100β dynamics, clarification of immune cell identity, and more critical discussion of limitations, particularly the use of only male rats and lack of pain behavior assessment. Overall, the study adds meaningful insight into Inco/A's potential role in nerve regeneration and warrants consideration after revision.
1. Consider making the title more specific, “Nerve-lesion model” is vague. “Chronic constriction injury (CCI) model in rats” would be more informative and precise.
2. The abstract lacks numerical data. Including specific values (e.g., % change in SFI, g-ratio improvement, CGRP reduction) would help convey the strength of the findings up front.
3. You chose a two-dose regimen (day 0 and day 21), but didn’t justify why this timing was selected. Please explain whether this is based on prior studies, pilot experiments, or clinical analogs.
4. You only performed histology at day 56. Including intermediate points (e.g., day 21 or 35) could help validate whether Inco/A accelerates or merely delays changes in regeneration markers like S100β and CGRP.
5. Only male rats were used, which the authors acknowledge as a limitation. However, this could be emphasized more strongly given the growing importance of sex differences in pain and regeneration research.
6. The reduction of S100β in Inco/A-treated rats is interpreted as “completion” of regeneration. But an alternative explanation could be premature Schwann cell inactivation. Consider additional markers (e.g., c-Jun, Ki67) to confirm SC status.
7. The reduction in leukocyte count is intriguing, but vague. It would be much stronger if the authors could identify whether these are macrophages, neutrophils, or lymphocytes. I suggest at least discussing potential immune subsets involved.
8. While the discussion mentions several prior studies, a table summarizing differences in BoNT/A formulation, timing, dose, and delivery (e.g., intraneural vs. perineural) would greatly help contextualize this work.
9. The manuscript emphasizes functional recovery but doesn’t present any direct nociceptive behavior assays (e.g., von Frey, thermal hyperalgesia). These would support the claims on anti-nociceptive effects via CGRP.
10. The justification for using Kruskal-Wallis vs. ANOVA isn’t always clear. For example, Figure 1c mixes these without explanation. Please clarify normality assumptions or show residual plots if possible.
11. Was the experimenter blinded to treatment groups during data collection and analysis? This information is important and missing from the Methods.
12. All authors are affiliated with Merz Therapeutics, which markets Inco/A. This should be more transparently addressed in the discussion when interpreting positive outcomes to avoid perceived bias.
13. The manuscript inconsistently uses terms like “neurotoxin,” “BoNT/A,” “Inco/A,” and “neurotoxin A” interchangeably. Standardizing terminology throughout would improve clarity.
14. Some figure panels (e.g., Figure 3a, CGRP and NF200) are difficult to interpret without higher magnification insets or contrast adjustments. Consider improving image presentation for better assessment.
15. The discussion lightly touches on potential human relevance. You might expand on how this perineural BoNT/A approach could be adapted for post-surgical nerve injury or chronic neuropathies in clinical settings.
Comments on the Quality of English LanguageThe quality of English throughout the manuscript is generally good. However, several sections would benefit from minor editing to improve sentence flow, remove redundancy, and ensure consistency in terminology (e.g., “BoNT/A” vs. “neurotoxin” vs. “Inco/A”).
Reviewer 3 Report
Comments and Suggestions for Authors
The manuscript submitted for review is an interesting scientific and research work concerning the influence of IncobotulinumtoxinA on the regenerative processes observed in damaged peripheral nerves. The work is written concisely and contains very interesting results. The authors have shown that repeated administration of the toxin to the area of ​​the damaged nerve significantly improves and accelerates the regenerative processes activated in the nerve, compared to a single injection of the neurotoxin. Additionally, the presented studies indicate for the first time that inhibiting the activity of pro-inflammatory factors has a great influence on the acceleration of regenerative processes and increases the possibility of complete recovery, at the same time indicating that repeated administration of the toxin significantly reduces the adverse effects of the immune system activated by pro-inflammatory factors, and consequently significantly accelerates the regenerative processes. The research results are presented in the work in a comprehensible and concise manner, the graphics attached to the work are well-chosen and significantly facilitate the understanding of the presented results. The research methods used in the work are correctly selected and allow for obtaining reliable data. The statistical methods used to analyze the obtained results are appropriately selected and correctly used. I strongly support the publication of this manuscript; however I recommend the minor revision of manuscript. Some small corrections should be made to the text according to the following comments:
Results
Figure 1 and 2 and 3- explain with full name all abbreviations that appear for the first time in the figures description
Materials and Methods
Although the number of animals used in the study is given in the description of the graphs, information on the number of animals used in the study should be included in the description of each research method used
References
no citation in the manuscript text of item 25 from the bibliography
Additionally, the bibliography must be prepared in accordance with the journal's requirements
Author Response
Comments 1:
Results
Figure 1 and 2 and 3- explain with full name all abbreviations that appear for the first time in the figures description
Response 1:
Thank you for this observation. We have now modified the figure legends and added full name to the abbreviations used as suggested.
Comments 2:
Materials and Methods
Although the number of animals used in the study is given in the description of the graphs, information on the number of animals used in the study should be included in the description of each research method used.
Response 2:
We appreciate the comment from the reviewer. In the whole study, we used 12 animals for group as indicated in the Behavioural tests section.
We have included the next sentence in the Animals section to make it clearer for the reader: “Unless specified, all animals underwent all assays described, and all readout values from these assays were included in the analysis.”
Comments 3:
References
no citation in the manuscript text of item 25 from the bibliography
Additionally, the bibliography must be prepared in accordance with the journal's requirements
Response 3:
Thank you for this good observation. All items were corrected and the format of the bibliography in the revised manuscript is now aligned with the journal’s requirements.
Round 2
Reviewer 2 Report
Comments and Suggestions for Authors
Although some revisions were made by the authors, the quality of this manuscript does not meet the journal's publication standard. The presentation of data, especially in imaging figures, is not clear and methodologically stringent. Additionally, the immunohistochemistry pictures utilized for sciatic nerve examining do not provide orientation (proximal vs. distal) specification and hence decrease interpretability. In addition, the quality of images presented is too low, and this hinders the clear identification of cellular and structural details. It is crucial to obtain higher resolution images with clear orientation and uniform scale bars.
Overall, the manuscript needs to have the data presentation and figure quality greatly improved before it can be accepted for publication. It would be optimal if the authors could thoroughly revise these aspects in the subsequent revision.
1. The IHC photographs of the sciatic nerve do not have absolute orientation labels. Proper interpretation requires that the "proximal" and "distal" directions with respect to the injury site be labeled in the images provided. This explanation is particularly critical for longitudinal sections, as directionality influences the biological interpretation of regeneration and marker distribution.
2. The IHC and histological images shown here (e.g., Figures 3–5) lack enough resolution, impeding the transparent observation of cellular and structural features. It would be advisable if the authors provided images with more resolution to support a comprehensive appraisal of the data quality and relevant interpretations by the readers and the reviewers.
3. Apart from resolution enhancement, it would be helpful if the authors provided insets or larger panels of critical regions to point out characteristic features, i.e., regenerating fibers, inflammatory cells, or Schwann cell markers.
4. Scale bars in some of the figures are not consistent or are missing in some figures. Make sure that all microscopic images include clearly labeled scale bars for reference.
5. It may be possible to enhance the study's transparency and reproducibility if the authors stated complete specifications regarding imaging parameters employed (e.g., objective magnification and pixel resolution) in the Methods section or in the figure legend.
Round 3
Reviewer 2 Report
Comments and Suggestions for Authors
In the legend to some figures (Figures 3-5, for example), it is written that "the right side represents the proximal area relative to the injury site, while the left side corresponds to the distal area." Such a statement conflicts with normal anatomical terminology: proximal is the direction that is closer to the body (e.g., spinal cord), whereas distal is the direction further away from the body (e.g., in the direction towards the paw). If you would rather denote locations with reference to the site of injury instead of the body anatomy, it is wise to state this deviation clearly. Also, the injury site should be well marked in the images to help substantiate your assertions. In the absence of these definitions, existing directional markers are confusing and unscientific.
I also have to raise serious issues about the credibility and interpretability of the results presented:
1. Both immunostaining and histological micrographs are shown as magnified fields, with no indication that the same anatomical regions were examined across experimental groups. This renders your comparisons suspect in consistency and validity.
2. In research pertaining to models of sciatic nerve injury, i.e., those with chronic constriction injury (CCI), it is essential that low-magnification, wide-field presentations are provided that clearly identify the area of injury. Showing only high-magnification images in isolation, without spatial reference, is not acceptable and diminishes the credibility of data presented.
In light of these major concerns, I cannot accept the manuscript in its present form. I would strongly suggest that the authors redraw the figures to show low-magnification overviews, clearly label the site of injury, and make image selection anatomically consistent and reproducible between all groups.